# Levels of SARS-CoV-2 antibodies among fully vaccinated individuals with Delta or Omicron variant breakthrough infections

Nina Breinholt Stærke [1,2,15] ✉, Joanne Reekie [3,15], Henrik Nielsen [4,5], Thomas Benfield [6,7], Lothar Wiese[8], Lene Surland Knudsen[8], Mette Brouw Iversen[8], Kasper Iversen[9], Kamille Fogh[9], Jacob Bodilsen[4,5], Maria Ruwald Juhl[4], Susan Olaf Lindvig[10], Anne Øvrehus[10], Lone Wulff Madsen[10,11], Vibeke Klastrup[1], Sidsel Dahl Andersen[1], Anna Karina Juhl[1], Signe Rode Andreasen[1], Sisse Rye Ostrowski [7,12], Christian Erikstrup [2,13], Thea K. Fischer [7,14], Martin Tolstrup[1,2], Lars Østergaard[1,2], Isik Somuncu Johansen[10,11], Jens Lundgren [3,7] & Ole Schmeltz Søgaard [1,2]

SARS-CoV-2 variants of concern have continuously evolved and may erode vaccine induced immunity. In this observational cohort study, we determine the risk of breakthrough infection in a fully vaccinated cohort. SARS-CoV-2 anti-spike IgG levels were measured before first SARS-CoV-2 vaccination and at day 21–28, 90 and 180, as well as after booster vaccination. Breakthrough infections were captured through the Danish National Microbiology database. incidence rate ratio (IRR) for breakthrough infection at time-updated anti-spike IgG levels was determined using Poisson regression. Among 6076 participants, 127 and 364 breakthrough infections due to Delta and Omicron variants were observed. IRR was 0.29 (95% CI 0.15–0.56) for breakthrough infection with the Delta variant, comparing the highest and lowest quintiles of anti-spike IgG. For Omicron, no significant differences in IRR were observed. These results suggest that quantitative level of anti-spike IgG have limited impact on the risk of breakthrough infection with Omicron.

Vaccination of the population is widely accepted as a critical element in controlling the COVID-19 pandemic[1,2]. Most COVID-19 vaccines were developed as two-dose regimens, but with the evidence of waning immunity over time after vaccination[3,4], and the emergence of new, more transmissible or immune escaping variants of concern[5–8], booster doses are now recommended in many countries. However, the optimal timing and target groups for COVID-19 vaccine booster doses are uncertain and further evidence is necessary to inform health authorities and policy makers in planning COVID-19 vaccination programs.

Level of anti-spike IgG has been associated with protection against SARS-CoV-2 infection after vaccination with the ChAdOx1, mRNA-1273 and BTN162b2 vaccines, as well as after natural infection[9–11]. In all cases the association has been described as a gradient, where the risk of SARS-CoV-2 infection is reduced with increasing levels of anti-spike IgG, rather than a binary cut-off model. However, these studies were performed before the emergence of the Omicron variant.

On 24th November 2021 the World Health Organization (WHO) was first notified about the new SARS-CoV-2 variant B.1.1.529

(Omicron) which had been discovered in South Africa. The Omicron variant has more than 30 mutations in the spike protein region, several of which are known from other variants to affect transmissibility and immune escaping capabilities[12–16]. Omicron spread quickly across the world and as of 28th December constituted 90% of the variant polymerase chain reaction (PCR) tested cases in Denmark[17]. Evidence suggests that effectiveness of the current SARS-CoV-2 vaccines against infection with the Omicron variant is reduced compared to other SARS-CoV-2 variants[6,7,18–21] and that protection gained from booster doses, wane over time[22]. Adaptation of vaccines or the addition of repeated booster doses to vaccine programs may be necessary to maintain high levels of protection against COVID-19 in the population[6,20–23].

The objective of this study was to determine the level of total SARS-CoV-2 anti-spike IgG in fully vaccinated participants experiencing breakthrough infection, and to investigate the risk of breakthrough infection at different levels of total anti-spike IgG stratified by SARS-CoV-2 variant. Furthermore, the frequency of severe COVID-19 disease was determined.

## Results

The study included 6076 participants with no documented SARS-CoV-2 infection prior to inclusion and at least one study visit recorded after baseline. The majority of participants ($n = 3939$, 64.8%) were censored at the date of their third vaccine dose and re-entered the analysis at their next study visit. A further 1811 participants (29.8%) were censored at the date of their third vaccine dose and did not re-enter the analysis, the remaining 326 participants (5.4%) did not receive a third dose during follow-up. The analysis included a total of 1,210,203 person-days of follow-up with each study participant followed for a median of 243 days (interquartile range [IQR] 202–272). SARS-CoV-2 breakthrough infections were documented in 504 individuals during the study period yielding an overall incidence rate (IR) of 0.42 per 1000 person days (95% CI 0.38–0.45). Of these 504 breakthrough infections, 364 were classified as the Omicron variant (IR 1.98, 95% CI 1.78–2.19), 127 cases as the Delta variant (IR 0.14 per 1000 person days, 95% CI 0.12–0.17) and 13 as unknown or other variants. Variant information is shown in Supplementary Table 1.

### Baseline characteristics

Participants experiencing breakthrough infection were younger with a median age of 56 years (IQR 47–68) compared with those without documented infection (median age 64 years, IQR 55–75). Among participants without documented breakthrough infection 15.8% (881/5572) did not have a PCR or antigen test during follow-up. The proportion of healthcare workers and participants receiving an AstraZeneca/mRNA vaccine regimen were higher in the group experiencing breakthrough infection (12.1% (61/504) and 10.5% (53/504), respectively) as compared to those without breakthrough infection (6.7% (371/5572) and 5.2% (291/5572), respectively). Of note, there was a clear association between being healthcare worker and receiving an AstraZeneca/mRNA regimen. For participants infected with the Omicron variant, the proportions of females and healthcare workers were higher than among those infected with the Delta variant (63.7% (232/364) vs 51.2% (65/127), and 15.1% (55/364) vs 1.6% (2/127)). Baseline characteristics of all participants stratified by breakthrough infection and variant are shown in Table 1. Kaplan–Meier plots show the time from baseline, and 28-day post-booster visit to breakthrough infection, overall and for each specific variant (see Supplementary Figs. 1 and 2).

### Antibody level and risk of breakthrough infection

We found a dose-dependent association between the level of total anti-spike IgG and the risk of breakthrough infection with the Delta variant after adjusting for age, sex, being healthcare worker, and SARS-CoV-2 transmission level at time of infection. Compared to participants with the lowest antibody levels ($\leq 1.77$ $\log_{10}$ BAU), those in the highest quintile of total anti-spike IgG ($>3.02$ $\log_{10}$ BAU) had a 71% lower rate of breakthrough infection with the Delta variant (incidence rate ratio) (IRR) 0.29, 95% CI: 0.15–0.56, $p = 0.0002$) and those with total anti-spike IgG between 2.69–3.02 $\log_{10}$ BAU had a 57% lower rate (IRR 0.43, 95%CI 0.25–0.75, $p = 0.003$). Lower IRRs were also observed in participants with total spike IgG between 2.29–2.69 $\log_{10}$ BAU, and 1.77–2.29 $\log_{10}$ BAU but the rates were not significantly different to those with the lowest levels ($<1.77$ $\log_{10}$ BAU). The same finding was observed when total anti-spike IgG was fitted as a continuous variable (IRR 0.72 per log10 increase in total anti-spike IgG BAU, 95% CI

**Table 1 | Participant demographics at baseline (14 days after second SARS-CoV-2 vaccination)**

| | Breakthrough infection during follow-up | | | Variant | |
|---|---|---|---|---|---|
| | Total ($N = 6076$) | No ($N = 5572$) | Yes ($N = 504$) | Delta ($N = 127$) | Omicron ($N = 364$) |
| Age at enrolment (median, IQR) | 64 (54, 75) | 64 (55, 75) | 56 (47, 68) | 58 (48, 69) | 55 (46, 68) |
| **Age Group (N, %)** | | | | | |
| <55 | 1615 (26.6) | 1380 (24.8) | 235 (46.6) | 51 (40.2) | 177 (48.6) |
| 55–64 | 1543 (25.4) | 1425 (25.6) | 118 (23.4) | 34 (26.8) | 79 (21.7) |
| ≥65 | 2918 (48.0) | 2767 (49.7) | 151 (30.0) | 42 (33.1) | 108 (29.7) |
| **Sex (N, %)** | | | | | |
| Male | 2675 (44.0) | 2474 (44.4) | 201 (39.9) | 62 (48.8) | 132 (36.3) |
| Female | 3401 (56.0) | 3098 (55.6) | 303 (60.1) | 65 (51.2) | 232 (63.7) |
| **Vaccine type (N, %)** | | | | | |
| Pfizer-BioNTech | 3369 (55.4) | 3113 (55.9) | 256 (50.8) | 72 (56.7) | 179 (49.2) |
| Moderna | 2363 (38.9) | 2168 (38.9) | 195 (38.7) | 54 (42.5) | 135 (37.1) |
| AstraZeneca/mRNA | 344 (5.7) | 291 (5.2) | 53 (10.5) | 1 (0.8) | 50 (13.7) |
| **Population group (N, %)** | | | | | |
| Individuals at increased risk[a] | 1434 (23.6) | 1322 (23.7) | 112 (22.2) | 28 (22.0) | 83 (22.8) |
| Healthcare worker | 432 (7.1) | 371 (6.7) | 61 (12.1) | 2 (1.6) | 55 (15.1) |
| General population | 4210 (69.3) | 3879 (69.6) | 331 (65.7) | 97 (76.4) | 226 (62.1) |

The Delta variant analysis includes 6063 participants, and the Omicron variant analysis includes 5050 participants.
[a]Individuals at increased risk includes cancer patients in active treatment, patients with immunodeficiencies (acquired or inherent), organ transplant recipients, hemodialysis patients, and patients with severe hematological, pulmonary or rheumatological diseases.

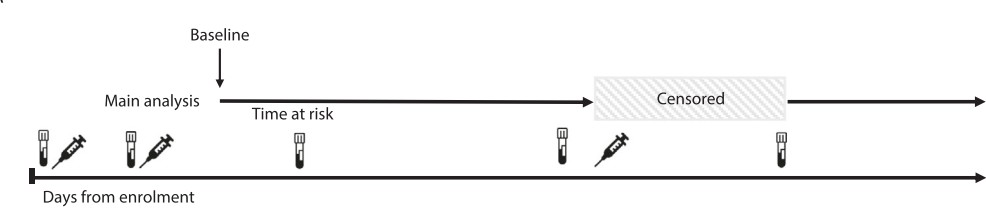

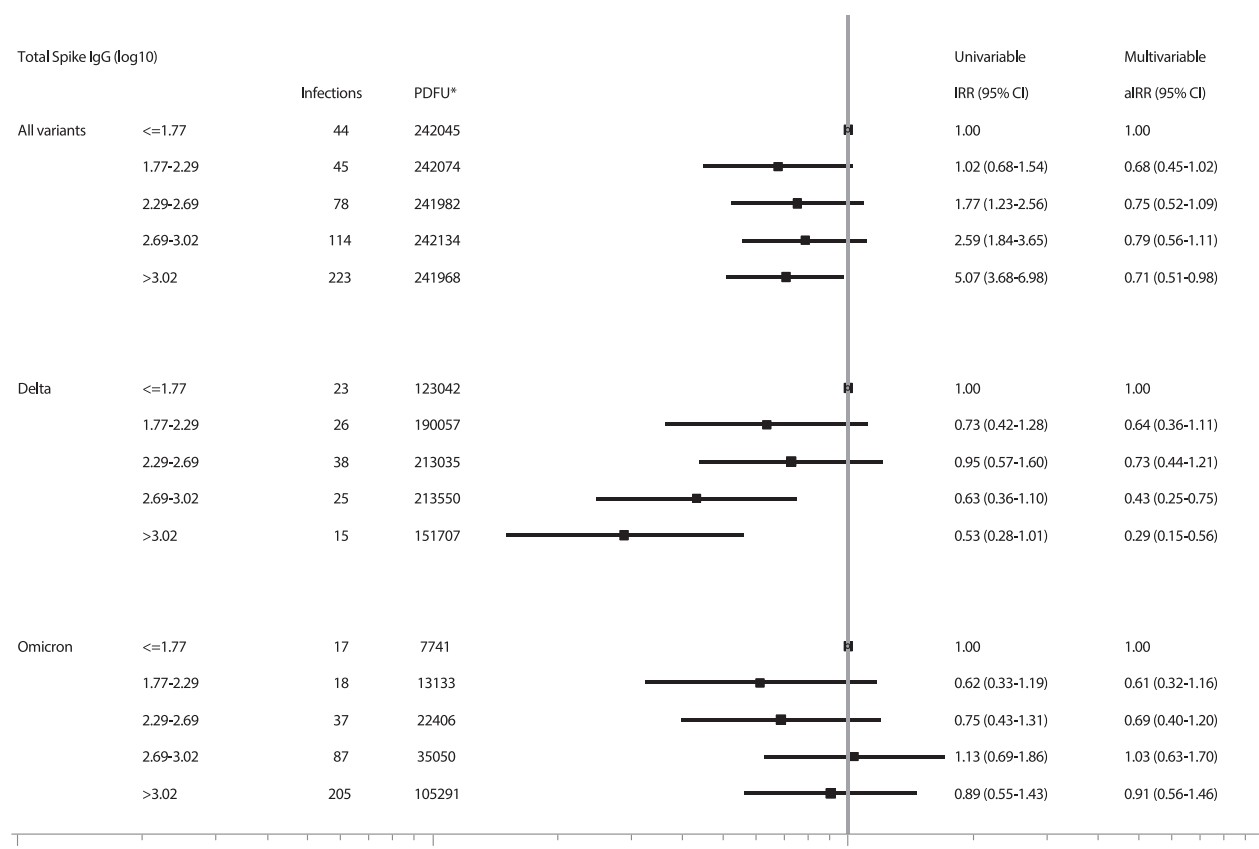

**Fig. 1 | Main analysis adjusted and unadjusted incidence rate ratios for breakthrough infection. A** Schematic overview of the analysis design. The syringe icon represents vaccinations, the blood sample icons represent blood draws and the shaded area represents the censored time period. Participants were censored at the time of third SARS-CoV-2 vaccination and re-entered the analysis at the time of the post-booster blood draw. **B** Forest plot of the adjusted incidence rate ratios (aIRR) for breakthrough infections. Incidence rate ratios (IRR) and adjusted incidence rate ratios (aIRR) for breakthrough infections calculated using a Poisson regression analysis for each quintile of SARS-CoV-2 anti-spike IgG log10 BAU, stratified by viral variant. The multivariable models were adjusted for age at enrolment (per year later), gender (male vs female), being healthcare worker (no vs yes) and transmission level with two-sided chi-squared tests for each variable in the model. The multivariable model for the Omicron variant did not include transmission level as all Omicron breakthrough infections occurred during the very high transmission period. All variants analysis: $n = 6076$, Delta analysis: $n = 6063$, Omicron analysis: $n = 5050$. The forest plot presents the aIRR and 95% confidence intervals from the multivariable models. *Person-days of follow-up.

0.60–0.84, $p < 0.001$). Conversely, we found no significant association between levels of total anti-spike IgG and the risk of breakthrough infection with the Omicron variant. The IRR of infection with the Omicron variant was 0.91 (95% CI 0.56–1.46) for participants in the highest quintile of IgG compared with those in the lowest quintile. IRR for breakthrough infections at each quintile of total anti-spike IgG are shown overall and stratified by variant in Fig. 1. The analysis was repeated for anti-receptor binding domain (RBD) antibodies with similar findings. Results are shown in Supplementary Fig. 3.

The median time from last antibody measurement to breakthrough infection was 34 days (IQR: 15–60 days) for all variants,

40 days (IQR 20–66 days) for breakthrough infection with delta, and slightly shorter (30 days IQR 14–56 days) for omicron. As antibody levels gradually decrease after vaccination, we performed a sensitivity analysis including only breakthrough infections occurring in the first 30 days after any serology measurement. This analysis included a total of 233 cases of breakthrough infection; 41 cases categorized as the Delta variant, 185 cases categorized as the Omicron variant and 7 cases with other or unknown variants. The sensitivity analysis was consistent with our primary findings. Participants with total anti-spike IgG > 2.69 log10 BAU had an IRR of 0.09 (95% CI 0.03–0.26) for breakthrough infection with the Delta variant, compared with those with total anti-

A

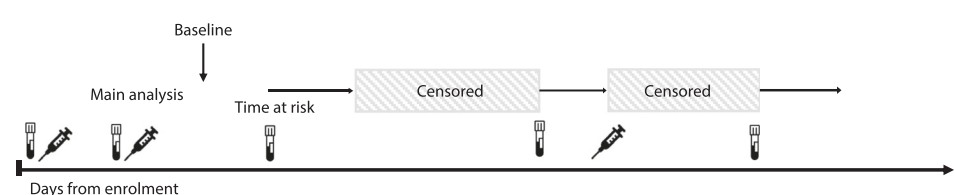

B

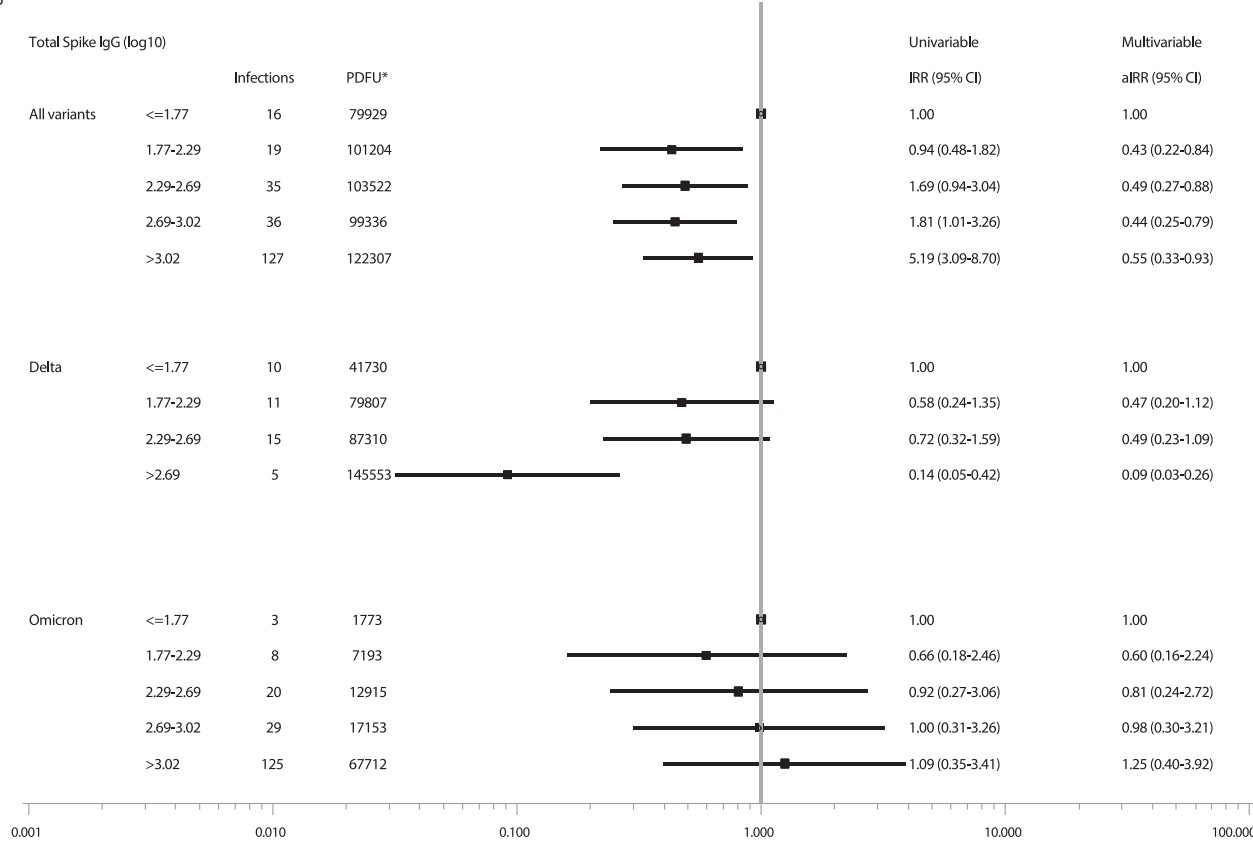

**Fig. 2 | Sensitivity analysis adjusted and unadjusted incidence rate ratios for breakthrough infection. A** Schematic overview of the analysis design. The syringe icon represents vaccinations, the blood sample icon represents blood draws, and the shaded areas represents the censored time periods. The analysis includes the first thirty days following each study visit blood draw. **B** Forest plot of the adjusted incidence rate ratios (aIRR) for breakthrough infections Incidence rate ratios (IRR) and adjusted incidence rate ratios (aIRR) for breakthrough infections calculated using a Poisson regression analysis for each quintile of SARS-CoV-2 anti-spike IgG log$_{10}$ BAU, stratified by viral variant. Multivariable models were adjusted for age at enrolment (per year later), gender (male vs female), being healthcare worker (no vs yes) and transmission level with two-sided chi-squared tests for each variable in the model. The multivariable model for the Omicron variant did not include transmission level as all infections occurred during the very high transmission period. The multivariable model for the Delta variant did not include healthcare worker, as there were no delta infections in this group in the sensitivity analysis. All variants analysis: $n = 6073$, Delta analysis: $n = 6034$, Omicron analysis: $n = 4388$. The forest plot presents the aIRR and 95% confidence intervals from the multivariable models. *Person days of follow-up.

spike IgG ≤ 1.77 log$_{10}$ BAU. Also, no significant association remained between the level of total anti-spike IgG and the risk of breakthrough infection with the Omicron variant (Fig. 2). Results of the sensitivity analysis using total anti-spike antibody levels in quintiles specific for each variant were consistent to the main analysis for the delta variant. For the omicron variant we observed a slightly increased risk of breakthrough infection at higher antibody levels (Supplementary Fig. 4) which was likely due to the timing of the booster vaccine doses coinciding with the peak of the omicron outbreak in Denmark.

### Factors associated with breakthrough infection
Increasing age was associated with a reduced risk of breakthrough infection in both the univariable and multivariable model (IRR: 0.97,

95% CI: 0.96–0.97 and IRR: 0.97, 95% CI: 0.97–0.98). Increasing transmission of SARS-CoV-2 in society was highly correlated to increasing risk of breakthrough infection in both the univariable and multivariable models. Female sex and being healthcare worker were both associated with the risk of breakthrough infection in the univariable model (IRR: 1.23, 95% CI: 1.03–1.46, and IRR:1.80, 95% CI: 1.40–2.32), however, in the multivariable model particularly after adjusting for age, this association disappeared (Table 2).

### Severe COVID-19 disease
Out of 504 breakthrough infections, one (0.2%) was identified as a case of severe COVID-19 disease with symptoms requiring hospital admission and medical treatment. In addition, 9/504 (1.8%) cases received

**Table 2 | Factors associated with breakthrough infection (n = 6076)**

| | Breakthrough infections | PDFU[a] | Univariable | | Multivariable | |
|---|---|---|---|---|---|---|
| | n | | IRR (95% CI) | p-value | aIRR (95% CI) | p-value |
| **Total spike quintiles (log 10)** | | | | | | |
| ≤1.77 | 44 | 242,045 | 1.00 | | 1.00 | |
| 1.77–2.29 | 45 | 242,074 | 1.02 (0.68–1.54) | 0.91 | 0.68 (0.45–1.02) | 0.06 |
| 2.29–2.69 | 78 | 241,982 | 1.77 (1.23–2.56) | 0.002 | 0.75 (0.52–1.09) | 0.13 |
| 2.69–3.02 | 114 | 242,134 | 2.59 (1.84–3.65) | <0.0001 | 0.79 (0.56–1.11) | 0.17 |
| >3.02 | 223 | 241,968 | 5.07(3.68–6.98) | <0.0001 | 0.71 (0.51–0.98) | 0.03 |
| **Age at enrollment (per 10 years older)** | | | 0.97 (0.96–0.97) | <0.0001 | 0.97 (0.97–0.98) | <0.0001 |
| **Sex** | | | | | | |
| Male | 201 | 543,681 | 1.00 | | 1.00 | |
| Female | 303 | 666,522 | 1.23 (1.03–1.46) | 0.02 | 0.98 (0.82–1.17) | 0.78 |
| **Healthcare worker** | | | | | | |
| No | 443 | 1,124,364 | 1.00 | | 1.00 | |
| Yes | 61 | 85,839 | 1.80 (1.40–2.32) | <0.0001 | 1.18 (0.89–1.55) | 0.24 |
| **Transmission level[b]** | | | | | | |
| Low | 7 | 310,364 | 1.00 | | 1.00 | |
| Intermediate | 36 | 569,908 | 2.60 (1.25–6.29) | 0.01 | 2.80 (1.25–6.30) | 0.01 |
| High | 51 | 134,818 | 16.77 (7.61–36.95) | <0.0001 | 16.56 (7.49–36.64) | <0.0001 |
| Very high | 410 | 195,113 | 93.17 (44.15–196.62) | <0.0001 | 94.49 (44.58–200.30) | <0.0001 |

Incidence rate ratios (IRR) and adjusted incidence rate ratios (aIRR) for breakthrough infections modelled in a univariable and multivariable poisson logistic regression model, with two-sided chi-squared tests for each variable in the model.
[a]Person-days of follow-up.
[b]Low: <10 cases/100,000 population/day, moderate: 10–40 cases/100,000 population/day, high: 41–85 cases/100,000 population/day and very high: >85 cases/100,000 population/day.

early out-patient treatment with monoclonal antibodies and/or remdesivir to reduce the risk of hospitalization and a severe disease course. For breakthrough infections with the Delta variant, 1 of 127 (0.8%) resulted in severe COVID-19, while 0 of 364 (0%) cases of infection with the Omicron variant caused severe disease.

## Discussion

The inverse relationship between the level of total anti-spike IgG and risk of breakthrough infection, observed for the Delta variant, demonstrates the protective effect by the SARS-CoV-2 vaccines against infection with the Delta variant. The same association was not observed for the Omicron variant, suggesting that quantitative level of anti-spike IgG have limited impact on the risk of breakthrough infection with Omicron. However, we observed only one case of severe COVID-19 disease among the Delta variant cases, and none among participants infected with the Omicron variant, suggesting a dissociation between anti-Spike antibody levels and the risk of progression to severe Omicron infection.

Our findings regarding the Delta variant are consistent with those of Wei et al., who described an association between anti-spike IgG levels and protection from SARS-CoV-2 infection with the Delta variant in a large sample of the UK population[11]. The findings of Dimeglio et al suggested that breakthrough infections with Omicron may occur at higher levels of total SARS-CoV-2 antibodies compared to infections with Delta[24], however, their study was not designed to calculate risk of infection at different levels of antibodies. Andrews et al. demonstrated that third doses of mRNA vaccines provided an increase in protection against breakthrough infections with Omicron in a test-negative case-control design[22]. In addition, Wratil et al. demonstrated in vitro that three exposures to SARS-CoV-2 spike protein, either by vaccination or infection, elicit a superior level of neutralization capacity against Omicron, compared to fewer exposures. However, the neutralization capacity against Omicron was still several fold lower than that for other variants[25]. The reduced neutralization capacity appears even more pronounced for the Omicron variant BA.2 than BA.1[26]. Of note, more

than 70% of SARS-CoV-2 infections in Denmark were caused by Omicron BA.2 from the second week of January 2022 and until the end of the study period[27]. The present study focuses on the quantitative level of anti-spike IgG, irrespective of the number of exposures to vaccines or infection, and it is possible that an association between the number of exposures to SARS-CoV-2 antigen, or neutralization capacity of the antibodies, and the risk of breakthrough infection with Omicron is present in our cohort.

Christensen et al. compared the characteristics of more than 20000 COVID-19 patients infected with either the alpha, Omicron or Delta variant[28]. They found a significantly higher proportion of fully vaccinated individuals among those infected with the Omicron variant, indicative of a reduced vaccine effect, which is in line with our results. Furthermore, they showed that Omicron caused proportionally fewer hospital admissions, similarly to data from South Africa, which demonstrated a decoupling of the rates of death and hospitalizations from the rate of new SARS-CoV-2 infections in the time following the emergence of Omicron, compared to the time before[29]. These findings indicate that omicron might cause less severe disease than other variants but pre-existing immunity to SARS-CoV-2 through vaccination and/or prior infection may also reduce COVID-19 severity. Notably, the ENFORCE cohort includes a large proportion of participants with comorbidities and of older age groups[30], and yet only 0.2% of all breakthrough infections resulted in severe COVID-19.

The finding that increasing age is associated with a reduced risk of breakthrough infection may seem counterintuitive, as increasing age is related to reduced vaccine efficacy due to immunosenescence. However, behavioural factors such as isolation, distancing, mask use and increased hand hygiene may also affect the risk of infection and the elderly may have displayed a more cautious behaviour. In addition, there was wide-spread SARS-CoV-2 transmission among school children and their parents in Denmark during the fall and winter of 2021/22, increasing the exposure to SARS-CoV-2 in the younger age groups[27]. Similarly, the univariate analysis of overall risk of breakthrough infection showed that participants belonging to the highest

quintile of antibodies had a 5.04 higher risk of breakthrough infection compared to those in the lowest quintile. However, when adjusting for age, sex, being healthcare worker and SARS-CoV-2 transmission level at the time of infection, and when stratifying for variant, this finding was reversed. Participants at risk of a reduced antibody response due to older age or concomitant diseases were probably also more likely to isolate themselves, and thereby reduce the risk of exposure, whereas younger or healthier persons were more likely to be exposed to the virus through everyday activities. The level of transmission in the Danish society varied greatly in the study period, and the highest levels of transmission were seen in December 2021 through February 2022, during which the majority of the cohort also received their third vaccination. This meant that the risk of exposure to the virus was very high at the same time as the participant's levels of antibodies were high, due to the third vaccination.

Our study also has some limitations. We included antibody data as time updated variables. This method entails a risk of over- or underestimation of the antibody levels at time of infection. To avert this effect, we censored participants from the time of booster vaccination until the date of the post booster blood sample. However, there may still be a risk of over- or underestimating the antibody levels at other time points. The participants were not randomized to the different vaccines used, as this was not feasible at the time of inclusion, because vaccine supply was limited. Vaccination groups differed in terms of age and comorbidities, as described elsewhere[30]. Therefore, we have not compared the risk of breakthrough infections between the different vaccine regimens. We did not have specific variant information in all cases and therefore we used the time periods where Delta and Omicron respectively were most prevalent to determine the variant. After 7th of January 2022 variant PCR testing of new SARS-CoV-2 cases was no longer performed in Denmark, since Omicron comprised more than 90% of cases. We may therefore have misclassified the specific SARS-CoV-2 variant in some instances. Breakthrough infection was captured through positive PCR tests. However, in the group without documented breakthrough infection 15.8% did not have a registered PCR test result during follow-up, and some may have had an undocumented SARS-CoV-2 infection. Finally, the current SARS-CoV-2 vaccines are all based on the Wuhan-Hu-1 SARS-CoV-2 strain and therefore we measured vaccine-induced antibody levels with a serology assay utilizing Wuhan-Hu-1 strain RBD and Spike antigens. Other variant-specific antibodies may correlate differently to the risk of breakthrough infection; however, investigating this was beyond the scope of the present study.

In conclusion, we found an association between higher levels of anti-spike antibodies and reduced risk of breakthrough infections for the Delta variant but could not demonstrate the same association for the Omicron variant. In addition, only one case of severe COVID-19 disease occurred in the cohort during follow-up.

## Methods

ENFORCE was designed as an open-label, non-randomized, parallel group, observational cohort study enrolling adults residing in Denmark before their first SARS-CoV-2 vaccination offered through the Danish COVID-19 vaccination program. Eligible participants were adults 18 years or older being offered a SARS-CoV-2 vaccination through the Danish vaccination program, who were willing to participate and had not previously received any SARS-CoV-2 vaccine. Inclusion took place at seven study sites (Aalborg, Silkeborg, Aarhus, Odense, Roskilde, Hvidovre and Herlev), covering all five Danish regions, from February to July 2021. The study was approved by the Danish Medicines Agency (Eudra CT number:2020-006003-42) and the Ethics Committee of the Central Denmark Region (#1-10-72-337-20). The study was registered on clinicaltrials.gov (NCT04760132). All participants received written and oral information about the study before providing their consent to provide blood samples and clinical

data to the study. The modified primary endpoint was the level of SARS-CoV-2 spike antibody-associated risk of breakthrough infection, which is reported in the present study. Secondary outcomes included: detailed immunological assessment in subgroups of participants of markers of cellular immunity, breakthrough infections by vaccine type throughout the 24-month follow-up period, local and systemic vaccine reactions within 14 days of vaccination by vaccine type, grade 3 and 4 adverse event and serious adverse events by vaccine type and grade 1 and 2 adverse events by vaccine type. These will be reported elsewhere.

### Baseline data
Study participants were vaccinated with either BTN162b2 (Pfizer-BioNTech), mRNA-1273 (Moderna), or one dose of ChAdOx1 (AstraZeneca) and subsequent dose/doses with a mRNA vaccine. The decision to vaccinate and vaccine type used was determined by routine care. Data on age, sex, SARS-CoV-2 vaccine, and vaccination date were collected at baseline. Date and type of SARS-CoV-2 vaccine were confirmed by cross referencing with the national Danish Vaccination Register (Statens Serum Institut, Copenhagen, Denmark). Data were collected using the REDCap electronic data capture tool (version 10.6.18) hosted at the Capital Region of Denmark.

### SARS-CoV-2 serology measurements
Blood draws were made for serology measurements at 14 to 0 days prior to the participants' first SARS-CoV-2 vaccination, at 5 to 0 days prior to the second vaccination, and at day 90 and day 180 after the first vaccine dose. Additional visits were scheduled prior to and 28 days after the participants received their third SARS-CoV-2 vaccination. Plasma levels of SARS-CoV-2 RBD and Spike directed IgG were measured at each study visit using a diagnostic multi-antigen serology assay (Meso Scale Diagnostics LLC, Maryland, United States of America) based on Wuhan-Hu-1 strain RBD and Spike antigens. The assay was performed at the Research Laboratory at the Department of Infectious Diseases, Aarhus University Hospital (Aarhus, Denmark). Antibody levels were converted from IU/mL to BAU/mL by a conversion factor of 0.00901 for SARS-CoV-2 Spike and 0.0272 for SARS CoV-2 RBD. In addition, SARS-CoV-2 Spike antibodies were measured on serum samples utilizing the WANTAI ELISA based assay (Beijing Wantai Biological Pharmacy Enterprise Co., Ltd., Beijing, China), which was performed at Statens Serum Institut (Copenhagen, Denmark).

### Breakthrough infections
SARS-CoV-2 breakthrough infection was defined as SARS-CoV-2 PCR positivity any time after 14 days since the second dose of any SARS-CoV-2 vaccine. Data on all SARS-CoV-2 test and virus characterization results, including viral variant, was extracted from the Danish national microbiology database MiBa (Statens Serum Institut, Copenhagen, Denmark) using the participants' unique civil registration numbers. Due to a huge increase in the number of infections in December 2021, routine variant determination for all SARS-CoV-2 isolates was stopped on 21 December 2021. Breakthrough infections without variant information were classified as Delta if they occurred between 1st of July and 1st of December 2021, where Delta was the predominant strain, and Omicron between 21st of December 2021 and 1st of February 2022, where Omicron was the predominant strain. Occurrence of variants in Denmark is provided as supplementary notes 1 and 2. Breakthrough infections occurring before 1st of July 2021, and those without variant information occurring between 1. and 21. of December 2021 were classified as unknown variant. Severe COVID-19 disease was defined as hospital admission due to symptomatic COVID-19 requiring medical treatment and/or oxygen supplement. Information about any hospital admission among study participants within -7 to 30 days of a positive SARS-CoV-2 PCR or antigen test was obtained from the Danish National Patient Register. Electronic medical records were examined to determine whether each case met the case definition for severe COVID-19 disease.

### SARS-CoV-2 transmission levels

The level of SARS-CoV-2 transmission in the Danish society during the study period were estimated using the daily reported number of new SARS-CoV-2 infections in Denmark (provided by Statens Serum Institut, Copenhagen, Denmark) to calculate a mean daily incidence for each week. Calendar weeks were then categorized into four transmission categories: low (<10 cases/100,000 population/day), moderate (10–40 cases/100000 population/day), high (41–85 cases/100,000 population/day) and very high (>85 cases/100,000 population/day).

### Statistical analysis

Baseline was defined as 14 days following the second dose of any SARS-CoV-2 vaccine. Participants with a positive SARS-CoV-2 PCR at any time prior to baseline, those positive for antibodies based on the Wantai Elisa Assay from the first study visit (prior to vaccination) and participants with no follow-up visits recorded after baseline were excluded from the analysis. A flowchart of exclusion from the study is provided as supplementary Table 2. Individuals were followed, until either first positive PCR test, withdrawal from the study, death, or 1st of February 2022. Participants who received a booster vaccination were censored at this date but re-entered the analysis at the date of their 28-day post-booster visit. Kaplan Meier plots show the time from baseline, and 28-day post-booster visit to breakthrough infection, overall and for each specific variant. Baseline demographics were stratified by the occurrence of breakthrough infection during follow-up and compared using chi-squared tests.

Overall crude incidence rates (IR) per 1000 person days of follow-up were calculated. Univariable and multivariable Poisson regression analysis was used to investigate the association between breakthrough infections and total anti-spike antibody levels (in quintiles). To calculate the IR and incidence rate ratios (IRRs), for variant-specific analyses follow-up was restricted to time periods when each variant was present in Denmark. Antibody levels were included as categorical time-updated variables with the last observation carried forward until a new measurement was available. Five categories were created based on the quintiles of the antibody levels in the cohort during the whole follow-up period. Potential confounding variables included in the multivariable model were determined a priori. The multivariable model, therefore included time-updated antibody level, age at enrolment (continuous, per year older), sex (Male, Female), being healthcare worker (No, Yes), and SARS-CoV-2 transmission level at the time of infection (Low, Intermediate, High, Very High), SARS-CoV-2 transmission level was included as a time updated variable. We performed two sensitivity analyses. The first restricted to breakthrough infections occurring during the first 30 days after blood sampling. The second used total anti-spike antibody levels in quintiles specific to the distribution of the antibody levels for the follow-up period for each variant analysis rather than for the whole follow-up period. Statistical analyses were performed using SAS Studio 3.8 on SAS 9.4 (SAS Institute Inc., Cary, NC, USA).

### Reporting summary

Further information on research design is available in the Nature Research Reporting Summary linked to this article.

## Data availability

The ENFORCE study is still ongoing. While the study is ongoing data and coding details may be made available to scientists only upon approval of an application sent to the ENFORCE Scientific Steering Committee and approval by relevant authorities. Applications for data must be sent to enforce.rigshospitalet@regionh.dk and will be handled within 6 weeks. Detailed information about data access may be found here: https://chip.dk/Research/Studies/ENFORCE/Study-Governance. Public study reports are available at https://chip.dk/Research/Studies/ENFORCE/Study-Reports.

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

## Acknowledgements
We thank the ENFORCE study group for their continued commitment to this study. A list of study group members is provided as supplementary notes. This study was fully funded by the Danish Ministry of Health (act 150, January 28th 2021, LØ).

## Author contributions
O.S., J.L., L.Ø., M.T., N.S. and J.R. conceptualized the work. T.B., L.W., H.N., I.J., L.K., M.I., K.I., K.F., J.B., M.J., S.L., N.S., A.Ø., L.M. and V.K. collected the data. M.T., S.D.A., A.J. and S.R.A. performed the laboratory analyses. J.L. and J.R. validated the data. C.E., T.F. and S.O. performed external supervision and mentorship. J.R. performed the data analysis and visualization. N.S., O.S. and J.R. wrote the initial manuscript. All authors reviewed the manuscript.

## Competing interests
H.N. declares participation on advisory board meeting with G.S.K. and M.S.D. T.B. declares receipt of unrestricted research or travel grants from GSK, Pfizer, Gilead Sciences, MSD; and being principal investigator on trials conducted by Boehringer Ingelheim, Roche, Novartis, Kancera, Pfizer, MSD and Gilead; Board member on Pentabase, and advisory board member for MSD, Gilead, Pfizer, GSK, Janssen and AstraZeneca; consulting fees from GSK and Pfizer; receiving donation of study drug from Eli Lilly; and receiving honorarium for lectures from GSK, Pfizer, Gilead Sciences, Boehringer Ingelheim, Abbvie and AstraZeneca. NS declares being principal investigator on studies conducted by Pfizer and Gilead. All other authors declare no competing interests.

## Additional information

[1]Department of Infectious Diseases, Aarhus University Hospital, Palle Juul-Jensens Boulevard 99, 8200 Aarhus N, Denmark. [2]Department of Clinical Medicine, Aarhus University, Palle Juul-Jensens Boulevard 82, 8200 Aarhus N, Denmark. [3]Center of Excellence for Health, Immunity and Infections, Rigshospitalet, University of Copenhagen, Blegdamsvej 9, 2100 Copenhagen, Denmark. [4]Department of Infectious Diseases, Aalborg University Hospital, Hobrovej 18, 9000 Aalborg, Denmark. [5]Department of Clinical Medicine, Aalborg University, Sdr. Skovvej 15, 9000 Aalborg, Denmark. [6]Department of Infectious Diseases, Copenhagen University Hospital—Amager and Hvidovre, Kettegård allé 30, 2650 Hvidovre, Denmark. [7]Departments of Clinical Medicine and Public Health, University of Copenhagen, Blegdamsvej 3B, 2200 Copenhagen, Denmark. [8]Department of Medicine, Zealand University Hospital, Sygehusvej 10, 4000 Roskilde, Denmark. [9]Department of Cardiology and Department of Emergency Medicine, Herlev-Gentofte Hospital, Borgmester Ib Juuls Vej 1, 2730 Herlev, Denmark. [10]Department of Infectious Diseases, Odense University Hospital, J. B. Winsløws Vej 4, 5000 Odense, Denmark. [11]Department of Clinical Research, University of Southern Denmark, J. B. Winsløws Vej 19.3, 5000 Odense C, Denmark. [12]Department of Clinical Immunology, Rigshospitalet, Tagensvej 20 2200, Copenhagen, Denmark. [13]Department of Clinical Immunology, Aarhus University Hospital, Palle Juul-Jensens Boulevard 99, 8200 Aarhus N, Denmark. [14]Department of Clinical Research, Nordsjællands University Hospital, Dyrehavevej 29, 3400 Hillerød, Denmark. [15]These authors contributed equally: Nina Breinholt Stærke, Joanne Reekie. ✉e-mail: ninase@rm.dk

