## [Peer Review File · Nature Communications]

Levels of SARS-CoV-2 antibodies among fully vaccinated individuals with Delta or Omicron variant breakthrough infectionsREVIEWER COMMENTS

Reviewer #1 (Remarks to the Author):

This is a well implemented longitudinal cohort study of breakthrough infections with the delta and omicron variants of SARS-CoV-2 in a large Danish population. The results show that anti-Spike IgG (presumably of ancestral variant) are associated with protection against breakthrough infection with the delta variant, but not the omicron variant. The results are in line with previous observations. However, the authors need to provide much more thorough descriptions of the serological assays and statistical methods.

Booster data.

Many of the study participants were given booster doses, but the data on numbers of booster doses has not been provided. It would be important to include summary data on booster doses, for example in Table 1.

Serological assays.

Additional description of the serological assays should be provided in the methods section. For example, only RBD, is mentioned. More importantly, there is no mention of the variant included the assay, presumably the ancestral strain. Due to the substantial differences between the ancestral and omicron RBDs, this is a very important limitation, and should be discussed in some detail. Although no significant associations were found between anti-Spike (ancestral?) IgG and breakthrough infection with omicron (Figure 1), this may not be true for anti-Spike (omicron).

Temporal data

An important aspect of the analysis that is lacking is a temporal visualization of the data. For example, the timings of vaccinations, boosters, delta infections and omicron infections throughout the study.

Statistical methods.

The statistical methods are not well described. A Poisson regression is an appropriate choice, but it would be key to provide details of the model and covariates used in equation form.

Interpretation of results

There are some very interesting results which would benefit from some additional interpretation from the authors. For example, comparing the lowest and highest quintiles, we see that individuals with high antibody levels have 5.04 times higher incidence than individuals with lower antibody levels – a really unexpected result. Stratification by variant reverses this association – suggesting something similar to Simpson's paradox. This also points us towards the imbalanced PDFU for omicron low quintile (7729) versus high quintile (105995) presumably due to the effect of booster doses. Note that everything in the multivariable analysis reads more or less as one would expect, but without a description of the model used for the multivariate analysis, it's not possible to interpret this.

Figure 3: Study design

I found this figure hard to follow. The legend doesn't really describe what is happening in the figure in an intuitive manner. What are the black squares?

Supplementary figures

Some of the supplementary figures are presented with descriptions in Danish. For non-Danish speakers, it would be helpful to have an English translation.

Access to data and code.

No statement is made regarding access to data or code.

Reviewer #2 (Remarks to the Author):

In an observational cohort study in Denmark, the authors of this manuscript investigated the relationship between the level of SARS-CoV-2 anti-spike IgG in fully vaccinated participants and the occurrence of breakthrough infection. The frequency of severe COVID-19 disease was also determined. While there was an association between higher levels of anti-spike IgG and reduced risk

of breakthrough infections with Delta variant, no association was found for the Omicron variant. Out of 504 breakthrough infections, there was only one case of severe COVID-19 during follow-up. Several studies showed an association between anti-spike IgG levels and protection against SARS-CoV-2 infection in pre-Omicron era but little is known regarding Omicron breakthrough infections. Therefore, this topic is important. However, this study has several limitations.

1. Anti-spike IgG levels were measured using the Meso Scale Diagnostics system. What is the linearity domain of this method? Results are expressed in IU and not in Binding Antibody Units / ml as recommended (Kristiansen Lancet 2021 ; Kumar Lancet 2021). This is very important for comparing results from one study to another.

Two immuno assays were used, one based in total spike and the other based on the receptor binding domain only. The findings were similar (line 108). Were quantitative levels obtained with the 2 methods correlated? Were the limit of detection and dynamic range identical?

2. At baseline, were the participants tested for previous SARS-CoV-2 infection? It is mentioned that there was no documented SARS-CoV-2 infection prior to inclusion (line 74). Many infections are symptomatic and are detected only by the presence of anti-nucleocapsid antibodies. A previous infection could have an impact on breakthrough infection.

3. Study outcome : SARS-CoV-2 breakthrough infection was defined as SARS-CoV-2 PCR positivity any time after 14 days since the second dose vaccine. What about asymptomatic infections?

4. Minor point :

- Table 1 : individuals at increased risk must be defined.

Reviewer #3 (Remarks to the Author):

This is an important manuscript, and the authors are to be commended for taking it on.

Clarifying the following comments would be helpful.

1) In the methods section, it states that the 1st blood draw was between 14 -0 days prior to the first SARS-CoV-2 vaccination. Were persons with positive antibodies at day 0 excluded from the study? The authors should clarify this.

2) The authors make an important point in stating univariate analysis was associated with increased risk of breakthrough infection in females and HCWs but this relationship disappeared with multivariable analysis. Can the authors explain what factors in the multivariable analysis changed the results?

3) How were the levels of transmission, mentioned in Table 2 defined? What is the difference, for example, in low versus intermediate transmission? It would be helpful to have this defined as a footnote to Table 2.

4) For Table 1, it would be helpful to add a footnote on the definition of persons at increased risk.

5) Was there any difference in time interval between the vaccination date and breakthrough infection date in the Delta era versus the Omicron era? Also, was there any difference in time interval between the date of serum measurement and breakthrough infection rates in the Delta and Omicron periods? In other words, could the lack of an association of antibody level breakthrough infection in the Omicron era be due to a longer time interval from the serum specimen used to assess antibody levels in the Omicron era?

6) This reviewer is not clear on what the differences in methodology were used in Figure 1 versus Figure 2. And are the bars for the multivariable analysis? This should be clarified.

7) Further, in Figures 1 and 2, the acronym PDFU is used. This is defined in the text but it would be helpful to add a footnote to the Figures, explaining what PDFU is.

8) What are the black boxes in Figure 3?

Reviewer #1 (Remarks to the Author):

This is a well implemented longitudinal cohort study of breakthrough infections with the delta and omicron variants of SARS-CoV-2 in a large Danish population. The results show that anti-Spike IgG (presumably of ancestral variant) are associated with protection against breakthrough infection with the delta variant, but not the omicron variant. The results are in line with previous observations. However, the authors need to provide much more thorough descriptions of the serological assays and statistical methods.

Q1: Booster data.

Many of the study participants were given booster doses, but the data on numbers of booster doses has not been provided. It would be important to include summary data on booster doses, for example in Table

A1: We agree with the reviewer that this is important information and have added details on the number of participants who received booster doses to the results section of the manuscript. (line 75-78)

“The majority of participants (n=3939, 64.8%) were censored at the date of their third vaccine dose and re-entered the analysis at their next study visit. A further 1811 participants (29.8%) were censored at the date of their third vaccine dose and did not re-enter the analysis, the remaining 326 participants (5.4%) did not receive a third dose during follow-up”

Q2: Serological assays.

Additional description of the serological assays should be provided in the methods section. For example, only RBD, is mentioned. More importantly, there is no mention of the variant included the assay, presumably the ancestral strain. Due to the substantial differences between the ancestral and omicron RBDs, this is a very important limitation, and should be discussed in some detail. Although no significant associations were found between anti-Spike (ancestral?) IgG and breakthrough infection with omicron (Figure 1), this may not be true for anti-Spike (omicron).

A2: Both RBD and Spike antibodies are measured in the multi-antigen serology assay, we apologise if this was not quite clear. The assay indeed uses ancestral strain (Wuhan-Hu-1) RBD and Spike antigen and we have added this information to the methods section (line 229-231).

“Plasma levels of SARS-CoV-2 RBD and Spike directed IgG were measured at each study visit using a diagnostic multi-antigen serology assay (Meso Scale Diagnostics LLC, Maryland, United States of America) based on the Wuhan-Hu-1 strain RBD and Spike antigens.”

The overall aim of the study was to assess the efficacy of the SARS-CoV-2 vaccines used in vaccination programmes, and as the vaccines are based on the Wuhan-Hu-1 strain we chose a serology assay targeting ancestral RBD and Spike antibodies. We agree with the reviewer that it may be so that higher levels of Omicron-specific anti-Spike antibodies would be protective against infection with the Omicron variant. However, we do not expect that the SARS-CoV-2 vaccines presently used would induce an Omicron-specific anti-spike antibody production. In addition, measuring antibodies against Omicron or variants of concern

would mean manually re-analyzing almost 30,000 plasma samples at a price of at least 2€ million, and therefore investigating this is beyond the scope of our study. We have added a comment about this topic to the discussion (line 202-205)

“Finally, the current SARS-CoV-2 vaccines are all based on the Wuhan-Hu-1 SARS-CoV-2 strain and therefore we measured vaccine induced antibody levels with a serology assay utilizing ancestral strain RBD and Spike antigens. Other variant-specific antibodies may correlate differently to the risk of breakthrough infection; however, investigating this was beyond the scope of the present study.”

Q3: Temporal data

An important aspect of the analysis that is lacking is a temporal visualization of the data. For example, the timings of vaccinations, boosters, delta infections and omicron infections throughout the study

A3: We agree this is an important point, we have included two Kaplan Meier plots to show 1) The time from baseline (two weeks after second vaccine dose) to breakthrough infection and 2) the time from the first study visit after their booster dose to breakthrough infection for those that received a booster dose and had a subsequent study visit.

We have referred to these in the methods section on line 269-270;

“Kaplan Meier plots were also produced to show the time from baseline, and 28-day post-booster visit to breakthrough infection, overall and for each specific variant.”

And in the results line 96-98;

“Kaplan Meier plots show the time from baseline, and 28-day post-booster visit to breakthrough infection, overall and for each specific variant (see supplementary materials).”

Q4: Statistical methods.

The statistical methods are not well described. A Poisson regression is an appropriate choice, but it would be key to provide details of the model and covariates used in equation form.

A4: The methods section has been updated and now includes a clearer description of the covariates that were included in the multivariable model and how they were fitted. We have not included the full equation for the Poisson regression model as it is not standard practice to provide such equations.

See lines 275-284;

“Univariable and multivariable Poisson regression analysis was used to investigate the association between breakthrough infections and total anti-spike antibody levels (in quintiles). To calculate the IR and incidence rate ratios (IRRs) for variant specific analyses, follow-up was restricted to time periods when each variant was present in Denmark. Antibody levels were included as categorical time-updated variables with the last observation carried forward until a new measurement was available. Five categories were created based on the quintiles of the antibody levels in the cohort during the whole follow-up period. Potential

confounding variables included in the multivariable model were determined a priori. The multivariable model, therefore included time-updated antibody level, age at enrolment (continuous, per year older), sex (Male, Female), being health care worker (HCW) (No, Yes), and SARS-CoV-2 transmission level at time of infection (Low, Intermediate, High, Very High). SARS-CoV-2 transmission level was included as a time updated variable.”

Q5: Interpretation of results

There are some very interesting results which would benefit from some additional interpretation from the authors. For example, comparing the lowest and highest quintiles, we see that individuals with high antibody levels have 5.04 times higher incidence than individuals with lower antibody levels – a really unexpected result. Stratification by variant reverses this association – suggesting something similar to Simpson’s paradox. This also points us towards the imbalanced PDFU for omicron low quintile (7729) versus high quintile (105995) presumably due to the effect of booster doses. Note that everything in the multivariable analysis reads more or less as one would expect, but without a description of the model used for the multivariate analysis, it’s not possible to interpret this.

A5: We appreciate the points made by the reviewer. We have included a more detailed description of the model used for the multivariate analysis in the methods section as outlined in the response to the reviewer above (A4). We agree that the unadjusted results reveal some interesting patterns, that we had not touched upon in the original manuscript. We believe that the association between higher antibody levels and breakthrough infection in the unadjusted model is caused both by differences in behavior, and temporal variations in the level of SARS-CoV-2 transmission. Participants at risk of poor vaccine responses due to older age or comorbidity were probably also more likely to isolate themselves, and thereby reduce the risk of exposure, whereas younger or healthier persons were more likely to be exposed to the virus through everyday activities and household transmissions. The level of transmission in the Danish society varied greatly in the study period, and the highest level of new infections were recorded in December through February, during which the majority of the cohort also received their third vaccination. This meant that the risk of exposure to the virus was very high at the same time as the participant’s levels of antibodies were high, due to the third vaccination. This paradox correlation is also why we adjusted for the level of transmission, as described in the methods section. We have added a section to the discussion addressing this point (line 179-188)

“Similarly, the univariate analysis of overall risk of breakthrough infection showed that participants belonging to the highest quintile of antibodies had a 5.04 higher risk of breakthrough infection compared to those in the lowest quintile. However, when adjusting for age, sex, being health care worker (HCW), and SARS-CoV-2 transmission level at time of infection, and when stratifying for variant, this finding was reversed. Participants at risk of a reduced antibody response due to older age or concomitant diseases were probably also more likely to isolate themselves, and thereby reduce the risk of exposure, whereas younger or healthier persons were more likely to be exposed to the virus through everyday activities. The level of transmission in the Danish society varied greatly in the study period, and the highest levels of transmission was seen in December 2021 through February 2022, during which the majority of the cohort also received their third vaccination. This meant that the risk of exposure to the virus was very high at the same time as the participant’s levels of antibodies were high, due to the third vaccination.”

Q6: Figure 3: Study design

I found this figure hard to follow. The legend doesn't really describe what is happening in the figure in an intuitive manner. What are the black squares?

A6: We thank the reviewer for this comment. We have removed figure 3 from the manuscript and instead added panels to figures 1 and 2 depicting the design.

Q7: Supplementary figures

Some of the supplementary figures are presented with descriptions in Danish. For non-Danish speakers, it would be helpful to have an English translation.

A7: This is indeed a very reasonable request and we have added English translations to the Danish descriptions that were missing.

“Supplementary materials. Delta variant occurrence in Denmark.

Development of Alpha and Delta SARS-CoV-2 variants in Denmark by calendar week. Bars show the weekly number of cases with either Alpha (yellow) or Delta (green) variants. The graphs show the weekly proportion of the two variants with 95% confidence intervals.”

Q8: Access to data and code.

No statement is made regarding access to data or code.

A8: Any data generated by the ENFORCE study may be made available to scientists upon approval of an application sent to the ENFORCE Scientific Steering Committee. This has been added to the end of the manuscript (line 289-291).

“Access to data and code

Data generated by the ENFORCE study and coding details may be made available to scientists upon approval of an application sent to the ENFORCE Scientific Steering Committee.”

Reviewer #2 (Remarks to the Author):

In an observational cohort study in Denmark, the authors of this manuscript investigated the relationship between the level of SARS-CoV-2 anti-spike IgG in fully vaccinated participants and the occurrence of breakthrough infection. The frequency of severe COVID-19 disease was also determined. While there was an association between higher levels of anti-spike IgG and reduced risk of breakthrough infections with Delta variant, no association was found for the Omicron variant. Out of 504 breakthrough infections, there was only one case of severe COVID-19 during follow-up.

Several studies showed an association between anti-spike IgG levels and protection against SARS-CoV-2 infection in pre-Omicron era but little is known regarding Omicron breakthrough infections. Therefore, this topic is important. However, this study has several limitations.

Q1: Anti-spike IgG levels were measured using the Meso Scale Diagnostics system. What is the linearity domain of this method? Results are expressed in IU and not in Binding Antibody Units / ml as recommended (Kristiansen Lancet 2021 ; Kumar Lancet 2021). This is very important for comparing results from one study to another. Two immuno assays were used, one based in total spike and the other based on the receptor binding domain only. The findings were similar (line 108). Were quantitative levels obtained with the 2 methods correlated? Were the limit of detection and dynamic range identical?

A1: This is a very valid point. We have converted all values to Binding Antibody Units as described by Meso Scale Diagnostics using the supplied conversion factor (line 232-233) to allow for cross analysis platform antibody comparisons.

As for the second point. The Meso Scale platform allow for antigen multiplexing and we used COVID-19 serology panel #2 for this work. This panel comprise Total Spike, the Receptor Binding Domain (RBD) and the Nucleocapsid as antigens. The quantitative measures of antibodies binding the two antigens Spike and RBD are highly correlated as expected. Levels of total IgG targeting the entire Spike proteins are higher than levels of antibodies targeting the RBD. This is also to be expected as Spike also comprise for instance the N-terminal domain (NTD) and the membrane proximal region. Importantly, RBD looks to be the most important region in terms of eliciting antibodies with neutralizing capacity.

The dynamic range is highly similar between Spike and RBD antigens both spanning 4-5 \log_{10} in linear dynamic range. Limit of detection is also on par between the two antigens.

Q2: At baseline, were the participants tested for previous SARS-CoV-2 infection? It is mentioned that there was no documented SARS-CoV-2 infection prior to inclusion (line 74). Many infections are symptomatic and are detected only by the presence of anti-nucleocapsid antibodies. A previous infection could have an impact on breakthrough infection.

A2: We assume that the reviewer meant "asymptomatic" and answered the question accordingly. We did measure anti-nucleocapsid antibodies throughout the study, however, as there is no established cut-off

value of anti-nucleocapsid antibodies for previous infection, we only excluded participants with a prior positive PCR test and those who were antibody positive based on the Wantai Elisa assay at their first study visit (-14 - 0 days prior to the first vaccine dose).

Q3: Study outcome: SARS-CoV-2 breakthrough infection was defined as SARS-CoV-2 PCR positivity any time after 14 days since the second dose vaccine. What about asymptomatic infections?

A3: As the reviewer suggests some participants may have been misclassified as not having breakthrough infections because they had an asymptomatic infection and therefore did not undergo PCR testing. The SARS-CoV-2 testing strategy in Denmark through the fall and winter 2021 and 2022 was quite rigorous and testing was recommended before social events, hospital appointment etc. irrespective of symptoms. However, 15.8% of the population without breakthrough infection did not have a registered PCR result during follow-up. This is mentioned in the limitations (line 200-202)

“Breakthrough infection was captured through positive PCR tests. However, in the group without documented breakthrough infection 15.8% did not have a registered PCR test result during follow-up, and some may have had an undocumented SARS-CoV-2 infection.”

Q4: Minor point:

- Table 1 : individuals at increased risk must be defined.

A4: We have added a definition of individuals at increased risk to table 1 as a footnote.

“Individuals at increased risk includes cancer patients in active treatment, patients with immunodeficiencies (acquired or inherent), organ transplant recipients, hemodialysis patients, and patients with severe hematological, pulmonary or rheumatological diseases.”

Reviewer #3 (Remarks to the Author):

This is an important manuscript, and the authors are to be commended for taking it on. Clarifying the following comments would be helpful.

Q1: In the methods section, it states that the 1st blood draw was between 14 -0 days prior to the first SARS-CoV-2 vaccination. Were persons with positive antibodies at day 0 excluded from the study? The authors should clarify this.

A1: Yes, we excluded participants with a previous positive SARS-CoV-2 PCR test from the present study and also participants who were positive for antibodies based on the Wantai Elisa Assay at enrolment (-14 - 0 days). This has been clarified in the methods (line 234-236 and 264-265)

“Additionally, SARS-CoV-2 Spike antibodies were measured on serum samples utilizing the WANTAI ELISA based assay (Beijing Wantai Biological Pharmacy Enterprise Co., Ltd., Beijing, China) which was performed at Statens Serum Institut (Copenhagen, Denmark).”

“Participants with a positive SARS-CoV-2 PCR at any time prior to baseline, those positive for antibodies based on the Wantai Elisa Assay from the first study visit (prior to vaccination), and participants with no follow-up visits recorded after baseline were excluded from the analysis.”

Q2: The authors make an important point in stating univariate analysis was associated with increased risk of breakthrough infection in females and HCWs but this relationship disappeared with multi-variable analysis. Can the authors explain what factors in the multivariable analysis changed the results?

A2: The inclusion of age in the multivariable model is the main reason for these factors becoming non-significant. As can be seen from both the univariable and multivariable models there is a very strong association with age, such that older individuals have a significantly lower risk of breakthrough infection aIRR 0.97 per year older, 95%CI 0.96-0.97. In our cohort the age distribution of males (median age 69 (IQR 57-77) vs. females (median age 61 (IQR52-72) and non-HCW (65, IQR 55-75) vs. HCW (49, IQR 35-59) were significantly different ($p < .0001$) which helps to explain why the association is seen in the univariable model but not after adjusting for age as a confounding variable. Please also see the answer A5 to reviewer #1.

The following has been added to the text (Line 132-133)

“Female sex and being HCW were both associated with risk of breakthrough infection in the univariable model (IRR: 1.23, 95% CI: 1.03–1.46, and IRR:1.80, 95% CI: 1.40–2.32), however, in the multivariable model particularly after adjusting for age, this association disappeared (table 2).”

Q3: How were the levels of transmission, mentioned in Table 2 defined? What is the difference, for

example, in low versus intermediate transmission? It would be helpful to have this defined as a footnote to Table 2.

A3: The levels of transmission are defined in the methods section line 254-259. However, we agree with the reviewer that it would be reasonable to include the definition as a footnote in table 2, and have done so.

“Low: <10 cases/100000 population/day, moderate: 10–40 cases/100000 population/day, high: 41–85 cases/100000 population/day and very high: >85 cases/100000 population/day.”

Q4: For Table 1, it would be helpful to add a footnote on the definition of persons at increased risk.

A4: We have added a definition of individuals at increased risk to table 1. See the reply A4 to reviewer #2.

Q5: Was there any difference in time interval between the vaccination date and breakthrough infection date in the Delta era versus the Omicron era? Also, was there any difference in time interval between the date of serum measurement and breakthrough infection rates in the Delta and Omicron periods? In other words, could the lack of an association of antibody level breakthrough infection in the Omicron era be due to a longer time interval from the serum specimen used to assess antibody levels in the Omicron era?

A5: Yes, there were differences in the time between most recent vaccine date and breakthrough infection. The majority of Delta infections occurred prior to receiving a booster dose of the vaccine, whereas the majority of the Omicron infections occurred shortly after receiving the booster dose. However, the time interval between the latest antibody measurement and breakthrough infection was actually slightly shorter in the Omicron time period compared to the Delta period. This information has been added to the results section (line 116-118). Please also see A4 to reviewer #1 regarding temporal data.

“The median time from last antibody measurement to breakthrough infection was 34 days (IQR: 15–60 days) for all variants, 40 days (IQR 20-66 days) for breakthrough infection with delta, and slightly shorter (30 days IQR 14-56 days) for omicron.”

Q6: This reviewer is not clear on what the differences in methodology were used in Figure 1 versus Figure 2. And are the bars for the multivariable analysis? This should be clarified.

A6: In figure 1 the primary analysis of incidence rate ratios for breakthrough infection is shown, whereas figure 2 shows the results of a sensitivity analysis where we only included breakthrough infections occurring in the 30 days following a study visit blood draw. For both figures the forest plots show the results of the multivariable analysis. We have changed the figure legends to make this clearer to readers. Additionally, we have changed the figures to include a visualization of the design used for the respective analyses.

“Figure 1:

a) Schematic overview of the analysis design. The syringe icon represents vaccinations, the blood sample icon represents blood draws, and the shaded area represents the censored time period. Participants were censored at the time of third SARS-CoV-2 vaccination and re-entered the analysis at the time of the post-booster blood draw. b) Primary analysis forest plot of the adjusted incidence rate ratios (aIRR) for breakthrough infections. Incidence rate ratios (IRR) and adjusted incidence rate ratios (aIRR) for breakthrough infections calculated using a Poisson regression analysis for each quintile of SARS-CoV-2 anti-spike IgG \log_{10} BAU, stratified by viral variant. The multivariable models were adjusted for age at enrolment (per year later), gender (male vs female), healthcare worker (no vs yes) and transmission level. The multivariable model for the Omicron variant did not include transmission level as all Omicron breakthrough infections occurred during the very high transmission period.

*Person-days of follow-up”

And,

a) Schematic overview of the analysis design. The syringe icon represents vaccinations, the blood sample icon represents blood draws, and the shaded areas represents the censored time periods. The analysis includes only the first thirty days following each study visit blood draw. b) Sensitivity analysis forest plot of the adjusted incidence rate ratios (aIRR) for breakthrough infections. Incidence rate ratios (IRR) and adjusted incidence rate ratios (aIRR) for breakthrough infections calculated using a Poisson regression analysis for each quintile of SARS-CoV-2 anti-spike IgG \log_{10} IU, stratified by viral variant. The forest plot depicts the results of the multivariable analysis. Multivariable models were adjusted for age at enrolment (per year later), gender (male vs female), healthcare worker (no vs yes) and transmission level. The multivariable model for the Omicron variant did not include transmission level as all infections were during the very high transmission period. The multivariable model for the Delta variant did not include healthcare worker, as there were no delta infections in this group in the sensitivity analysis.

*Person days of follow-up

Q7: Further, in Figures 1 and 2, the acronym PDFU is used. This is defined in the text but it would be helpful to add a footnote to the Figures, explaining what PDFU is.

A7: We agree on this and have added a footnote to each of the figures to clarify that PDFU stands for person days of follow-up. See also the reply (A6) above.

Q8: What are the black boxes in Figure 3?

A8: The black boxes in figure 3 were not intentional, and we apologize for the confusion caused by that. We have removed figure 3 from the manuscript. Instead, we have added panels in figure 1 and 2 schematically describing the design of the two analyses and hope that this is easier to interpret.

REVIEWER COMMENTS

Reviewer #1 (Remarks to the Author):

I am happy with the authors response to my previous comments, and believe the manuscript is now substantially stronger.

As a side note, unrelated to this manuscript, I would suggest the authorship team seek better value for money for their serological tests. At least EUR 2m for 30,000 serological tests is very bad value for money.

Reviewer #2 (Remarks to the Author):

Several points, including values expressed in Binding Antibody Units, have been addressed by the authors. However, one important point require attention (Reviewer 2, Q1 and A1). As illustrated in Figure I B (new version of the manuscript), the number of individuals per quintile is very heterogeneous for Omicron ($n = 105291$ for Total Spike IgG > 3.02 log versus $n = 7741$ for Total Spike IgG ≤ 1.77 log). To analyse the relationship between the level of SARS-CoV-2 antibodies and the occurrence of breakthrough infections a distinction between a Delta infection and an Omicron infection is necessary. Therefore Anti-Spike or RBD SARS-CoV-2 antibody quintiles must be calculated for each variant and not globally. In addition, the variable « variant (Delta/Omicron) » (Table 2) should be added in the multivariate model.

Reviewer #3 (Remarks to the Author):

This is an important manuscript and the authors have addressed well this reviewer's prior comments to them.

REVIEWER COMMENTS

Reviewer #2 (Remarks to the Author):

Q1. Several points, including values expressed in Binding Antibody Units, have been addressed by the authors.

However, one important point requires attention (Reviewer 2, Q1 and A1). As illustrated in Figure I B (new version of the manuscript), the number of individuals per quintile is very heterogeneous for Omicron ($n = 105291$ for Total Spike IgG > 3.02 log versus $n = 7741$ for Total Spike IgG ≤ 1.77 log). To analyse the relationship between the level of SARS-CoV-2 antibodies and the occurrence of breakthrough infections a distinction between a Delta infection and an Omicron infection is necessary. Therefore Anti-Spike or RBD SARS-CoV-2 antibody quintiles must be calculated for each variant and not globally.

A1: The level of transmission in the Danish society varied greatly in the study period, and the highest level of new infections were recorded in December through February, during which the majority of the cohort also received their third vaccination. This coincided with when the Omicron variant became dominant in Denmark resulting in the heterogeneous groupings for the antibodies that the reviewer highlights.

The quintiles used for the main analysis and shown in the main figures were derived from the cohort during the whole follow-up period. We have opted to keep the data presented this way so that the reader can easily compare the risk of breakthrough infection from the two different variants across the same antibody levels. Further when the total anti-spike antibody levels were fitted as a continuous variable we observed the same significant association of a lower risk of breakthrough infections from the delta variant with increasing antibody levels (aOR 0.72 per log₁₀ increase, 95%CI 0.60-0.85, $p=0.0001$) and no significant association with omicron infections (aOR 1.07 per log₁₀ increase, 95% CI 0.86-1.34). We also feel that this stratification makes it easier for the reader to see the clear heterogeneity in the antibody levels during the two different time periods when each of the variants were dominant in Denmark.

However, we agree with the reviewer that due to the unbalanced groupings particularly during the omicron period it is important to also include an analysis with Anti-Spike SARS-CoV-2 quintiles specific to each of the variants and have added this as a sensitivity analysis. For the delta variant the results are very consistent, and we observe the same decrease in risk with increasing antibody levels.

For the omicron variant, we observe the opposite and see a slight increased risk of breakthrough infection with omicron at higher antibody levels. As noted previously, this is very likely due to the timing of the booster vaccines coinciding with increasing prevalence omicron in society.

The following has been added to the manuscript and a figure is included in the supplementary material:

Methods (Lines 285-289)

“We performed two sensitivity analyses. The first restricted to breakthrough infections occurring during the first 30 days after blood sampling. The second used total anti-spike antibody levels in quintiles specific to the distribution of the antibody levels for the follow-up period for each variant rather than for the whole follow-up period.”

Results (Lines 125-129)

“Results of the sensitivity analysis using total anti-spike antibody levels in quintiles specific for each variant were consistent to the main analysis for the delta variant. For the omicron variant we observed a slightly increased risk of breakthrough infection at higher antibody levels (Supplementary figure 4) which was likely due to the timing of the booster vaccine doses coinciding with the peak of the omicron outbreak in Denmark.”

Q2. In addition, the variable « variant (Delta/Omicron) » (Table 2) should be added in the multivariate model.

A2: Thank you for this comment. Table 2 shows the results of the Poisson Regression model investigating factors associated with the risk of experiencing a breakthrough infection. The analysis population for this model includes all participants included in the study, not just those who had a breakthrough infection. As the variant type only applies to those who experience a breakthrough infection, we are unfortunately unable to include this additional variable in the multivariable model.